# A ten-year review of neonatal tetanus cases managed at a tertiary health facility in a resource poor setting: The trend, management challenges and outcome

Ezra Olatunde Ogundare[1,2]*, Adebukola Bidemi Ajite[1,2], Adewuyi Temidayo Adeniyi[1,2], Adefunke Olarinre Babatola[1,2], Adekunle Bamidele Taiwo[2], Odunayo Adebukola Fatunla[3], Alfred Airemionkhale[2], Oluwapelumi Adeyosola Odeyemi[2], Oladele Simeon Olatunya[1,2], Oyeku Akibu Oyelami[4]

1 Department of Paediatrics and Child Health, Ekiti State University, Ado-Ekiti, Nigeria, 2 Department of Paediatrics and Child Health, Ekiti State University Teaching Hospital, Ado Ekiti, Nigeria, 3 Department of Paediatrics, Afe Babalola University, Ado-Ekiti, Nigeria, 4 Department of Paediatrics and Child Health, Obafemi Awolowo University, Ile-Ife, Nigeria

* ogundarezra@gmail.com, tundeyogundare@yahoo.com

**Data Availability Statement:** All relevant data are within the manuscript and its Supporting Information files.

## Abstract

### Background

Neonatal Tetanus (NNT) is a vaccine preventable disease of public health importance. It is still being encountered in clinical practice largely in developing countries including Nigeria.

NNT results from unhygienic delivery practices and some harmful traditional cord care practices.

The easiest, quickest and most cost-effective preventive measure against NNT is vaccination of the pregnant women with the tetanus toxoid (TT) vaccine. The case-fatality rate from tetanus in resource-constrained settings can be close to 100% but can be reduced to 50% if access to basic medical care with adequate number of experienced staff is available.

### Materials and methods

This retrospective study reviewed the admissions into the Special Care Baby Unit (SCBU) of the Ekiti State University Teaching Hospital, Ado-Ekiti from January 2011 to December 2020. The folders were retrieved from the records department of the hospital; Information obtained from folders were entered into a designed proforma for the study.

### Results

During the study period, NNT constituted 0.34% of all neonatal admissions with case fatality rate of 52.6%. Seven [36.8%] of the babies were delivered at Mission home/Traditional Birth Attendant's place while 5 [26.3%] were delivered in private hospitals. Cord care was with hot water compress in most of these babies16 [48.5%] while only 9% of the mothers cleaned the cord with methylated spirit. Age at presentation of less than one week was significantly associated with mortality, same with presence of autonomic dysfunction. Low family socio-

**Funding:** The authors received no specific funding for this work.

**Competing interests:** The authors have declared that no competing interests exist.

economic class 5 was significantly associated with poor outcome, so also maternal age above 24 years.

## Conclusion

This study revealed that neonatal tetanus is still being seen in our clinical practice with poor outcome and the risk factors are the same as of old.

Increased public health campaign, promotion of clean deliveries, safe cord care practices, affordable and accessible health care provision are recommended to combat NNT scourge.

### Author summary

Neonatal Tetanus (NNT) is a vaccine preventable disease of public health importance still being encountered in clinical practice largely in developing countries however, there are still foci of tetanus cases in the developed world, although exceedingly rare. Tetanus is a toxin mediated disease responsible for the death of hundreds of newborns every year, unfortunately most of these deaths are not recorded because most of the affected babies do not have contact with health facilities. Only 5% of cases are reported to present in health facilities. The Maternal and Neonatal Tetanus Elimination initiative has made remarkable progress as more than three quarter of the priority countries have attained elimination, and the remaining countries seems to be making steady progress over the years. NNT is still being seen in our health facilities. Poor antenatal clinic attendance, unsupervised deliveries, poor cord care, low maternal literacy level and low socio-economic status were some of the major risk factors for NNT in this study. Mortality from NNT is still high as there is no means of ventilatory support for affected babies. There is the need to encourage mothers to attend antenatal care during pregnancy, get vaccinated and have supervised deliveries. There is also the need for public enlightenment campaign on the appropriate and approved cord care methods. It may also be important to consider incorporating tetanus vaccination into the School Health Programme (SHP) to ensure that every female child gets adequate protection before commencement of childbearing.

## Introduction

Neonatal Tetanus (NNT) is a vaccine preventable disease of public health importance still being encountered in clinical practice largely in developing countries including Nigeria. However, there are still foci of tetanus cases in the developed world, howbeit exceedingly rare [1]. Neonatal tetanus is a toxin-mediated disease which usually present with inability to suck in a newborn who has been sucking before, excessive cry or irritability, with or without fever, generalized body stiffness or rigidity and painful muscle spasms.

In approximately 20% of tetanus cases generally there is no obvious portal of entry [2]. However, for most cases of neonatal tetanus, the portal of entry is the umbilical stump. Ear piercing and infected circumcision sites are other known sources of the infection.

NNT results from unhygienic delivery practices and some harmful traditional cord care practices which include cutting the cord with non-sterile equipment such as razor blade;

application of shea butter, mentholatum containing balms and animal dung to the cord [3]. Hot compress is also applied to the cord stump in some cases.

Majority of the affected babies in Nigeria are delivered either at home, faith (mission) based homes, with untrained or poorly trained traditional birth attendants as only 43% of deliveries are attended by skilled birth attendant as revealed by the 2018 Nigeria Demographic and Health Survey (NDHS) report [4].

The easiest, quickest, and most cost-effective preventive measure against NNT is vaccination of the pregnant women with the tetanus toxoid (TT) vaccine [5]. Three doses of the TT vaccine given to a pregnant woman can provide a 5-year protection for the mother and the children at an average cost of about US$1.80 a decade ago [5]. Unfortunately, most of the mothers of the affected babies do not receive vaccination before or during pregnancy while some will only take one dose of tetanus toxoid vaccination in pregnancy [4].

The case-fatality rate from tetanus in resource-constrained settings can be close to 100% but can be reduced to 50% if access to basic medical care with experienced staff is available [6] and appropriate facilities.

Generally, morbidity and mortality from NNT occur in very young infants usually within the first 7 to 14 days of life [5].

Globally, NNT now accounts for about 1% of neonatal deaths, with a decline from 14% in 1993 [7,8] while in Nigeria, NNT accounts for about 20% of neonatal death [9]. Efforts aimed at worldwide tetanus elimination including NNT with neonatal tetanus elimination defined as less than one case of NNT per 1000 live births per district [10] has not yielded the desired result as different target dates have been missed. One major strategy developed by the World Health Organization [WHO] and its partners aimed at achieving this goal is improving vaccination among females before or during pregnancy and promotion of clean delivery using the Maternal and Neonatal Tetanus Elimination (MNTE) initiative [11].

Strategies involved in the MNTE initiative include: immunization of women especially pregnant women; Supplementary immunization activities in selected high-risk areas; Promotion of clean deliveries and clean cord care practices; and Reliable neonatal tetanus surveillance [11]. The MNTE strategy resulted in significant 88% reduction in NNT-related neonatal mortality [11]. Only 12 countries have not yet attained the Maternal and Neonatal Tetanus Elimination (MNTE) status as of July 2019 [11]. Nigeria is one of these twelve countries although there has been some progress. NNT cases are still seen in our health facilities although the numbers have reduced [12]. This may probably be due to some gains of the implementation of the MNTE strategies, or it could be due to under reporting as most NNT cases do not make it to the health facilities where they could be documented. NNT is under reported in Nigeria with only about 5% of cases reporting to health facilities [13]. Studies have indicated that only 2–5% of NNT cases are reported, and this makes planning and assessment of elimination programs difficult [14].

The burden of maternal and neonatal tetanus (MNT) also known as "silent killer" has been described as a health equity issue affecting mostly the underserved and disadvantaged population who also lack schools, employment opportunities and basic infrastructures, such as roads, health care services and communication [5].

A case of maternal and/or neonatal tetanus has also been described as a triple failure of the public health system–failure of; the routine immunization programme, antenatal care, and ensuring clean and safe delivery practices as well as clean cord care practices [15]. In Europe and North America NNT became increasingly rare through hygienic childbirth practices and cord care even before availability of tetanus vaccine [16,17]. According to the WHO hygienic delivery and cord care practices may be summarized as "six cleans" [18]; these include: clean hands; clean perineum, clean delivery surface; clean cord cutting, clean cord tying and clean

cord care [19]. Thus, it is imperative to promote clean deliveries and cord care practices in developing countries generally, Nigeria inclusive, so as to achieve NNT elimination by the year 2030 as projected.

Although the incidence of tetanus is declining truly, however NNT is still a major public health problem in developing countries and NNT cases are still being encountered in newborn of unimmunized or immunized mothers throughout the world but especially in developing countries [10]. There is need to draw the attention of policy makers to the task ahead if Nigeria truly desires to join the rest of the world in meeting the target date of eliminating NNT by the year 2030.

This study aims to describe the trend of NNT in a tertiary institution in Southwest Nigeria over the past decade, and to highlight the management challenges as well as areas for intervention for NNT elimination.

## Materials and methods

### Ethical statement

Ethical approval for the study was given by the Research and Ethics committee of the Ekiti State University Teaching Hospital, Ado-Ekiti, Ekiti State with approval number EKSUTH/A67/2021/06/004. Formal verbal consent was obtained from the Parents/Guardians of the study participants.

This retrospective study reviewed the admissions into the Special Care Baby Unit [SCBU] of the Ekiti State University Teaching Hospital (EKSUTH), Ado-Ekiti from January 2011 to December 2020. The EKSUTH is a tertiary public health facility providing health care to citizens of Ekiti State. It serves as a referral center to other hospitals within the state and other adjoining states like Osun, Ondo, Kwara, and Kogi that share borders with Ekiti State. The hospital is in Ado Ekiti which doubles as both the headquarters of Ado Local Government Area and the state capital. The city is mainly populated by the Yorubas of the southwestern part of Nigeria and has a population of approximately 313,690 inhabitants [20]. Agriculture is the main occupation of the people of Ekiti, and it is the major source of income for many in the state, while the women engage in trading. Agriculture provides income and employment for 75% of the population of Ekiti State. There are also civil servants, artisans, and small-business owners in Ekiti, and the minimum wage for the civil servants is like that of other states in Nigeria [21,22].

The hospital is a tertiary health facility, with a 16-bedded neonatal unit; 12 beds serve the inborn section while the outborn section has 4 beds. The SCBU is run by one consultant paediatrician, a senior registrar, one registrar, two house officers and fourteen nurses. The SCBU has facilities for neonatal resuscitation, phototherapy, and incubators. Neonates are admitted to the unit directly from the labour ward or the labour ward theatre as inborn patients while patients delivered outside the hospital are admitted into the outborn section.

### Data collection

All cases of neonatal tetanus admitted into the SCBU of Ekiti State University Teaching Hospital (EKSUTH) during the ten-year period, from January 2011 to December 2020, were reviewed. The case note numbers of the patients were retrieved from the admission and discharge register on the ward. The folders were retrieved from the records department of the hospital. Information obtained from folders were entered into a designed proforma for the study. The information obtained included: the patient's personal data, pregnancy and birth history, mother's antenatal care and immunization history, place of delivery, cord care practices, age at admission, age at onset of first symptom, period of onset, interval between first

symptom and presentation at the hospital, mother's level of formal education, socio-economic class of the family, home treatment offered, duration of admission and outcome.

Appropriate cord care was defined as the use of methylated only or use of chlorhexidine gel. The period of onset was defined as the interval in days between cessation of sucking and occurrence of spasms. Socioeconomic class was defined by the criteria described by Oyedeji et al [23].

Diagnosis of tetanus was made clinically according to the WHO diagnostic criteria [24] with all 3 of the following:

- A child who has been crying and feeding normally in the first two days of life.

- Onset of illness between day 3 and day 28 of life.

- Inability to suck [trismus], followed by generalized stiffness (muscle rigidity) with or without muscle spasms.

Diagnosis of associated Sepsis was made by positive blood culture result and or use of the World Health Organization (WHO) identified clinical signs suggestive of sepsis [25] difficulty feeding, convulsions, movement only when stimulated, respiratory rate >60 per min, severe chest in-drawing and axillary temperature >37.5˚C or <35.5˚C.

Diagnosis of autonomic dysfunction was made based on presence of tachycardia or bradycardia, arrhythmias, hyperpyrexia, and sweating.

## Treatment protocol of neonatal tetanus at the study centre

All cases of tetanus are admitted into the quiet section of the neonatal ward to reduce external stimuli. They all receive intravenous anti-tetanus serum at 10,000 IU within the first 24 to 48 hours of admission and intravenous metronidazole as the antibiotic of choice. Spasms are controlled with a combination of chlorpromazine, phenobarbitone and diazepam initially via intravenous route but this is later changed to oral medications via a nasogastric tube. The combination of the sedatives/muscle relaxants is dependent on the severity of the symptoms. The babies are fed with expressed breast milk via a nasogastric tube. A spasm chart is kept, vital signs are monitored before administration of sedatives and the doses of the sedatives adjusted accordingly depending on whether the spasms are increasing or reducing.

The patients are worked up for sepsis which include blood culture, umbilical wound swab for microscopy, culture and sensitivity, complete blood count and urine microscopy culture and sensitivity are done for patients appropriately.

Wound care is usually by wound debridement, application of hydrogen peroxide and subsequent cleaning with methylated spirit.

Outcome of managed cases are classified as discharged, discharged against medical advice (DAMA) and death.

## Data handling and analysis

The data obtained were analyzed using IBM SPSS version 25. The results were cross tabulated as frequency tables; means, standard deviations, percentages, and ranges was used as appropriate to describe continuous variables.

Test of associations were assessed using Chi-square, and a p-value of 0.05 or less was considered significant.

## Result

During the ten-year study period (2011–2020), a total of 5522 babies were managed in the Special Care Baby Unit (SCBU) out of which 19 babies were managed for neonatal tetanus (NNT)

thus, NNT constituted 0.34% of all neonatal admissions. Twelve [63.2%] were males and 7 (36.8%) were females giving a M:F ratio of 1.7: 1. The mean (SD) age of the babies at presentation was 8.79 ±5.05 days, 9 [47%] of them presented within the first week of life.

Seven [36.8%] of the babies were delivered at Mission home/Traditional Birth Attendant's place while 5 [26.3%] were delivered in private hospitals. Cord care was with hot water compress in most of these babies16 [48.5%], about 7 [20%] had mentholatum applied to the cord while only 1 [5.2%] mother claimed to clean the cord with only methylated spirit. The socio-demographic features of the babies are shown in Table 1 below. The umbilicus was the suspected/identified portal of entry of the tetanus infection in all the babies [100%].

The average maternal age was 24 years, 3 of the mothers were teenagers, 5 of the mothers were aged between 20 and 24 years, the ages of 6 of them were not documented. Five [26.3%] of the mothers were primiparous women. About a quarter of the mothers had no formal education, none had tertiary education, none belonged to social class 1 or 2 while 7[36.8%] were from low social class V, close to 60% of the mothers had no antenatal care during pregnancy and close to 60% of them did not receive tetanus toxoid vaccination during pregnancy (Table 2).

About a third of the babies presented with inability to suck or spasms while about a quarter of them had fever at presentation. Interval between the first symptom and the first spasm [period of onset] was less than 24 hours in 60% of the patients, 5 [26.3%] of the babies had low blood sugar (hypoglycaemia) at presentation while 7 [36.8%] of them had anaemia at presentation. About 50% of the patients were deemed to have features suggestive of sepsis at presentation while 3 [10.3%] had features of autonomic dysfunction at admission. More than half of the babies had fever while on admission while one [5.3%] had hyperpyrexia. Ten [52.6%] of the babies were given anti-tetanus serum during their admission. About half [52.6%] of the patients were on admission for more than 7 days (Table 3).

The highest incidence was observed in year 2011, with zero incidences recorded in years 2012 and 2015. However, in the last five years, the annual incidence hovers around 3 to 4% (Fig 1).

## Outcome of admission for NNT

More than half [52.6%] of the patients died, while 5 [26.3%] were discharged (Fig 2).

Table 1. Demographic characteristics of the babies.

| Variable | | Frequency (%) |
|---|---|---|
| Age of baby at presentation | 0–7 days | 9 (47.4) |
| | 8–14 days | 8 (42.1) |
| | ≥15 days | 2 (10.5) |
| Gender | Male | 12 (63.2) |
| | Female | 7 (36.8) |
| Place of Delivery | Home | 6 (31.6) |
| | Mission/TBAs | 7 (36.8) |
| | Private hospital | 5 (26.3) |
| | Not documented | 1 (5.3) |
| *Cord care | Shea butter | 6 (18.2) |
| | Hot water compress | 16 (48.5) |
| | Application of mentholatum | 7 (21.2) |
| | Local portion | 1 (3.0) |
| | Methylated spirit[β] | 3 (9.1) |

*—Multiple methods were used by some patients

β–Just 1 baby had only methylated spirit for cord care

**Table 2. Demographic characteristics of the mothers.**

| Variable | | Frequency (%) |
|---|---|---|
| Maternal age | Teenagers | 3 (15.8) |
| | 20 to 24 | 5 (26.3) |
| | 25 to 29 | 2 (10.5) |
| | 30 to 34 | 2 (10.5) |
| | 35 and above | 1 (5.3) |
| | Not documented | 6 (31.6) |
| Social-Economic Status (SEC) | Class I | 0 (0.0) |
| | Class II | 0 (0.0) |
| | Class III | 4 (21.1) |
| | Class IV | 6 (31.6) |
| | Class V | 7 (36.8) |
| | Not documented | 2 (10.5) |
| Mother's Highest level of Education | No education | 5 (26.3) |
| | Primary | 6 (31.6) |
| | Secondary | 8 (42.1) |
| | Tertiary | 0 (0.0) |
| Parity | Primiparous | 5 (26.3) |
| | Multipara | 9 (47.4) |
| | Grandmultipara | 1 (5.2) |
| | Not documented | 4 (21.1) |
| Antenatal Clinic Attendance | No | 11 (57.9) |
| | Yes | 5 (26.3) |
| | Not documented | 3 (15.8) |
| Tetanus Toxoid Vaccination | Nil | 11 (57.9) |
| | Not completed | 5 (26.3) |
| | Completed | 0 (0.0) |
| | Not documented | 3 (15.8) |

## Association between some selected variables and outcome of admission

Table 4 shows the association between some selected variables and outcome of admission. Age at presentation of less than one week was significantly associated with mortality, same with presence of autonomic dysfunction in the babies. Low family socio-economic class 5 was significantly associated with poor outcome, so also maternal age above 24 years. Administration of anti-tetanus serum to the patients while on admission was not significantly associated with a good outcome, same goes for the mothers' vaccination status during pregnancy.

## Discussion

This study set out to review the admissions of babies with diagnosis of tetanus over a ten-year period [2011–2020] at the Special Care Baby Unit (SCBU) of Ekiti State University Teaching Hospital (EKSUTH), Ado-Ekiti. The study made attempt at assessing the progress made towards elimination of neonatal tetanus using hospital-based data. The prevalence of NNT in this study was 0.34% which is less than 0.7% reported in 2011 by Onalo et al [26] in Zaria, 1% reported in 2015 by Mbarie and Abhulimhen-Iyoha in Benin City [27] and far less than 4.7% reported in 2012 by Peterside et al [12] in Bayelsa State, all in Nigeria. Emordi et al [28] in 2011 reported a prevalence of 0.4% of the total Paediatric admissions over a ten-year period in Enugu. No case was reported in the United Kingdom from 1984 to 2000 [29]; India has achieved the goal of neonatal tetanus elimination since mid-2015. The reason for this wide variation in prevalence may probably be due to the differences in the duration of the study conducted in different centres and the time interval between the previous studies and this current one, the minimum being about 5 years. Another possible reason could be the result of efforts and strategies targeted at achieving tetanus elimination in Nigeria. It could also be because

**Table 3. Presenting symptoms and other features.**

| Variable | | Frequency (%) |
|---|---|---|
| Period of onset | Less than 24 hours | 12 (63.2) |
| | Greater than 24 hours | 2 (10.5) |
| | Not documented | 5 (26.3) |
| Presenting complaint | Fever | 11 (23.9) |
| | Excessive cry | 3 (6.5) |
| | Inability to suck | 15 (32.6) |
| | Spasm | 15 (32.6) |
| | Apnoea | 1 (2.2) |
| | Difficulty in breathing | 1 (2.2) |
| Co-morbidities/Complication | Anaemia | 7 (24.1) |
| | Autonomic dysfunction | 3 (10.3) |
| | Hypoglycaemia | 5 17.2) |
| | Sepsis | 14 (48.3) |
| Packed Cell Volume | Normal | 3 (15.7) |
| | Low | 7 (36.8) |
| | High | 0 (0.0) |
| | Not documented | 9 (47.4) |
| Random Blood Glucose | Normal | 6 (31.6) |
| | Low | 5 (26.3) |
| | High | 0 (0.0) |
| | Not documented | 8 (42.1) |
| Temperature | Normal | 2 (10.5) |
| | Fever | 10 (52.6) |
| | Hyperpyrexia | 1 (5.3) |
| | Not documented | 6 (31.4) |
| Received Anti-Tetanus Serum | Yes | 10 (52.6) |
| | No | 3 (15.8) |
| | Not Documented | 6 (31.6) |
| Duration of admission | 1–7 days | 9 (47.4) |
| | 8–14 days | 3 (15.8) |
| | >15 days | 7 (36.8) |

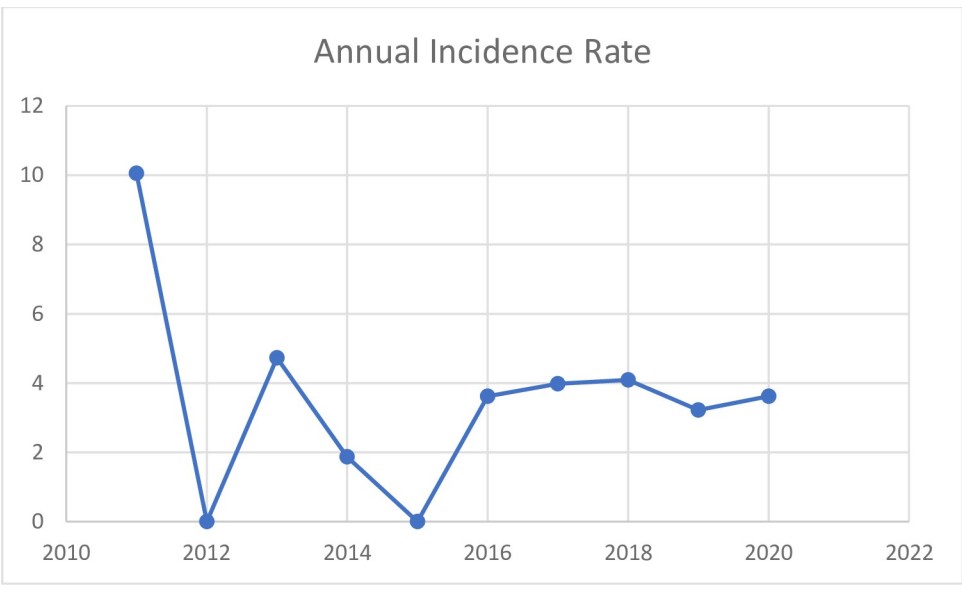

**Fig 1. Annual incidence rates of NNT.**

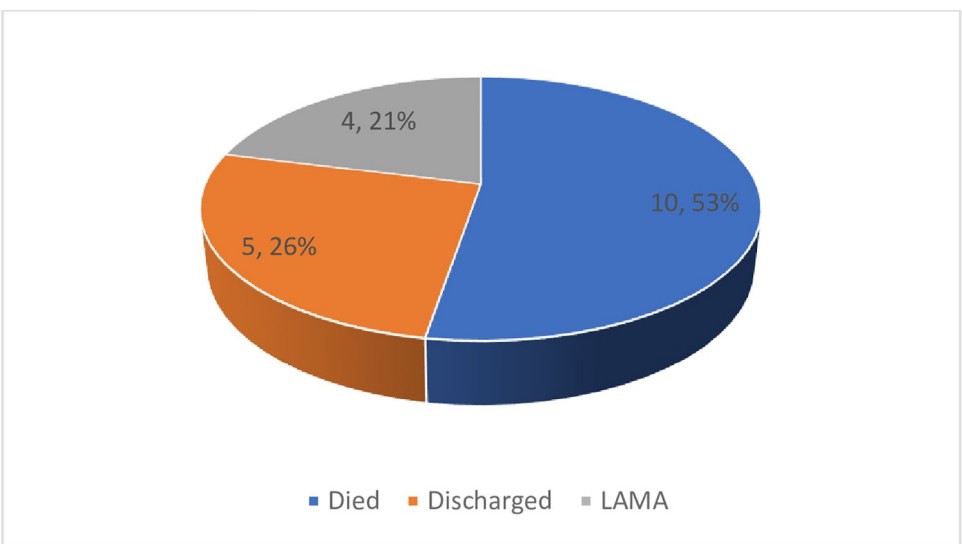

**Fig 2. Outcome of admission.**

**Table 4. Association between selected variables and outcome of care.**

| Variable | | Death | Discharge | LAMA | $\chi^2$ | p-value |
|---|---|---|---|---|---|---|
| Age Group | Within a week | 7 | 0 | 2 | 16.6 | |
| | More than a week | 3 | 5 | 2 | | **0.038** |
| Gender | Male | 6 | 2 | 4 | 10.26 | |
| | Female | 4 | 3 | 0 | | 0.171 |
| Co-morbidities/Complication | Anaemia | 0 | 0 | 2 | | |
| | Autonomic dysfunction | 3 | 0 | 0 | 6.6 | **0.015** |
| | Hypoglycaemia | 0 | 0 | 1 | | |
| | Sepsis | 7 | 5 | 1 | | |
| Cord Care | Appropriate$^\alpha$ | 1 | 0 | 0 | 14.3 | 0.733 |
| | Inappropriate | 7 | 4 | 4 | | |
| ATS administration | Yes | 5 | 2 | 3 | | |
| | No | 2 | 0 | 1 | 4.1 | 0.378 |
| | Not documented | 3 | 3 | 0 | | |
| Social economic status | Class I | 0 | 0 | 0 | | |
| | Class II | 0 | 0 | 0 | 5.6 | **0.023** |
| | Class III | 1 | 3 | 0 | | |
| | Class IV | 4 | 2 | 0 | | |
| | Class V | 3 | 0 | 4 | | |
| Maternal age | Teenagers | 0 | 0 | 3 | | |
| | 20 to 24 | 3 | 2 | 1 | 3.5 | **0.027** |
| | Greater than 24 | 3 | 1 | 0 | | |
| | Not documented | 4 | 2 | 0 | | |
| Maternal Antenatal Clinic Attendance | Yes | 4 | 1 | 0 | | |
| | No | 4 | 3 | 4 | | 0.352 |
| Maternal tetanus toxoid vaccination | Nil | 4 | 4 | 3 | | |
| | Completed | 0 | 0 | 0 | | 0.405 |
| | Not completed | 4 | 0 | 1 | | |
| | Not documented | 2 | 1 | 0 | | |

**α**—Appropriate cord care–cord care with only methylated spirit or the use of chlorhexidine gel

most of the affected neonates were not brought to the hospital for treatment as it's been previously reported that only 5% of cases are seen in health facilities [13].

The male preponderance observed in this study is in tandem with findings from most of the previous studies [26,30–32]. The exact reason for the male preponderance is not known but it may be due to the supposed premium placed on male children in the African setting hence, their being brought more for care compared to their female counterparts. The study by Mbarie and Abhulimhen-Iyoha [27] in Benin City and Peterside et al [12] in Bayelsa however reported female predominance.

About 50% of the babies presented within the first week of life suggesting that most of the patients managed had short incubation period which falls within the traditional incubation period of 3 days to 3 weeks. About 70% of the babies were delivered outside health facilities most likely without skilled birth attendants and under unhygienic conditions which agrees with the 2018 NDHS [4] report. It agrees with the findings from previous Nigerian studies [12,26,33] and studies in other countries, Turkey [34] and Pakistan [35]. As much as 30% of the babies were delivered at home most likely under unhygienic condition too. These figures show that there is a need for the government and policy makers to do more to encourage delivery in health care facilities such as making delivery at health facilities affordable, attractive, accessible, and less stressful to the pregnant women. About 30% of these babies were delivered in private hospitals and they still ended up with tetanus. The quality of care and hygienic practices at such hospitals may be contributory factors. This calls for adequate and proper monitoring of private hospitals and maternity homes to ensure strict compliance with hygiene.

Of all the patients in this study, only 1 (5.2%) claimed to have used only methylated spirit for the care of her baby's umbilical cord, about 50% used hot water compress as cord care method and about 20% each applied shea butter or mentholatum to the cord stump. This finding reflects the poor cord care practices that is still prevalent in our environment which is not different from the findings of previous studies [12,26–30,33,36]. This finding underscores the need to educate women of reproductive age groups and the adolescents on proper and approved cord care practices. The umbilicus was the probable portal of entry of the Clostridium tetani in all our patients. This shows the need to encourage mothers to use the appropriate cord care methods.

About a quarter of the mothers in this review had no formal education while none of the mothers in this study had tertiary education, also about two thirds of the families of babies affected by NNT in this study were from lower socio-economic classes IV and V. These findings confirm association between NNT and poverty as well as low educational level [5,9]. Majority (60%) of the mothers in this study had no antenatal care during pregnancy which agrees with the findings of previous studies in Nigeria [12,26,36–38]. This further affirms the role of antenatal care in the management of pregnant women in ensuring good outcome of both mother and baby. Female education is pivotal to child survival. An educated female/woman will probably be empowered economically and be able to take appropriate decision with respect to attending antenatal clinic, ensuring vaccination, choice of where to deliver her baby and adhering to simple hygiene instructions on cord care [3]. Female education impacts on a female's health seeking behaviour and decision making. With more than 70% of the mothers of babies with NNT in this study having primary and secondary education, the school health program offers an opportunity for a school-based vaccination programme which could commence during the elementary education and continued into secondary education, this could help in ensuring that most of the girl child have the 5 doses of vaccination before they commence childbearing [3,31].

About half of the babies had features of sepsis at presentation, this supports the findings of a previous study by Oyelami et al [39] in Ilesa. This probably is related to the cord care

practices as about 50% used hot compress and 20% each apply shea butter and mentholatum to the cord stump; these are practices that can encourage infection of the cord stump. This could also be due to delivery that took place in unhygienic environments. There is need for increased awareness about NNT and appropriate childcare practices among women of reproductive age group.

The trend in admission rate showed a significant drop after the year 2011 but it has maintained a plateau since the year 2016. There is a need to do more in terms of health education on basic hygiene practices after delivery in addition to improving vaccination indices if Nigeria hopes to eradicate NNT by the year 2030 according to SDG goal 3.

The case fatality rate for NNT in this study was 52.6% which is comparable to some previous reports from Nigeria [26,30,33,37,38] Pakistan [35] and Turkey [40] but it is higher than what some other studies in Nigeria [12,36,37] have reported. This high mortality is probably related to the fact that NNT has a poorer prognosis in most developing countries due to non-availability of facilities and quality medications needed to manage the disease. The use of intravenous magnesium sulphate and intrathecal antitoxin administration as spasm control methods is being considered as a possible way of avoiding the need for ventilatory support which if not available worsens mortality from NNT [1]. As observed in this study, all the three patients with features of autonomic dysfunction died. These babies might have benefitted from ventilatory support which is not available in the study location. These observations highlight some of the challenges in managing neonatal tetanus in resource poor settings. The high mortality rate recorded may also be due to the factor of age at presentation as most of our patients presented within the first seven days of life and it is well known that the shorter the incubation period, the higher the mortality rate [41]. The short incubation period may be due to high load of the tetanus toxin and its rapid distribution/spread in the neonate due to the naivety of their immune system or the virulence of the infecting agent [31]. The high case fatality rate recorded may also be due to the accompanying sepsis in our patients.

Age at admission of less than 7 days, features of autonomic dysfunction and factors such as low socio-economic class and maternal age above 24 years were all associated with poor outcome. The exact reason for the influence of maternal age above 24 years on outcome is not particularly known, however, it may be due to the belief that such women have in their ability to care for a newborn baby since they probably have had babies previously.

Antenatal clinic attendance (ANC) in this study was poor as less than 30% of the mothers had antenatal clinic attendance. This is far below the national average of 57% which is quite low [4]. This low ANC attendance will not afford mothers opportunity to be advised appropriately on the importance of supervised delivery and appropriate cord care methods as most of these babies were delivered outside health facilities with attended risk of unapproved and unhygienic traditional practices. The low ANC attendance can also result in lack of tetanus toxoid vaccination during pregnancy or incomplete tetanus toxoid vaccination status of the mothers. The poor ANC attendance as observed among these mothers might contribute to the continued occurrence of NNT in the study locality. Maternal tetanus toxoid vaccination was also not significantly associated with survival in this study unlike reports from some previous studies, this probably may be because none of the mothers in this study completed the two doses of vaccination in pregnancy. Two doses of TT in pregnancy have been reported to reduce NNT mortality by 94% [42], the current TT2 coverage in Nigeria is 40% which is less than the recommended 80% needed for coverage [43], while in Ekiti State the percentage of women whose last child birth was protected from tetanus was 88.2% [4]. The trend of Tetanus Toxoid Containing vaccine (TTCV2+) administration to pregnant women in Nigeria has not improved remarkably over the past few years with the rate ranging from 40% in 2015 to 62% in 2018 and down to 40% in 2019 [43]. The national average rate of Protection at birth (PAB)

coverage for newborns against tetanus is 60% which is far lower than the recommended 80% needed for adequate protection [43]. Administration of anti-tetanus serum to the patients did not improve survival too. This could be due to the challenge of getting the anti-tetanus serum of good quality because of poor preservation by the outlets where most patients get it to buy. There is the need for appropriate regulation of the sales of such important medication.

To reduce deaths from NNT requires reducing the prevalence of the disease and this can be achieved through increasing the vaccination coverage of pregnant women and women of reproductive age group. Another measure is to encourage pregnant women to have supervised delivery, this will require making health care facilities more friendly, accessible, affordable, and less stressful for the populace. There is also the need to train and retrain the traditional Birth Attendants (TBAs) on the importance of clean deliveries and cord care practices.

The contribution of NNT to neonatal admissions in this study is quite low compared to previous reports from Nigeria. There is still need for increased surveillance, case notification and reporting of NNT cases so that Nigeria can move towards elimination of NNT.

There is also the need for strong political will with respect to having appropriate policies that will ensure all the facilities needed for NNT prevention and treatment viz-a-viz well-equipped health care facilities, cold chain, trained and well-motivated personnel, as well as the vaccine/medications are all available.

## Conclusion

This study reveals that neonatal tetanus is still being seen in our clinical practice and that the mortality is still high due to lack of facilities for its proper management. There is need for increased public health campaign to help achieve the elimination drive and increased investment in health care delivery to improve the survival of affected babies.

## Recommendation

There is also the need to incorporate tetanus vaccination into the SHP to increase the number of females who get vaccinated before commencement of childbearing.

## Supporting information

**S1 Data. The neonatal tetanus data.**
(XLSX)

**S2 Data. The data for the figures.**
(XLSX)

## Acknowledgments

We appreciate the health information management officer who assisted in retrieving the case notes of the patients and all other colleagues who contributed to the success of this study.

## Author Contributions

**Conceptualization:** Ezra Olatunde Ogundare, Adebukola Bidemi Ajite.

**Data curation:** Ezra Olatunde Ogundare, Adekunle Bamidele Taiwo, Alfred Airemionkhale, Oluwapelumi Adeyosola Odeyemi.

**Formal analysis:** Ezra Olatunde Ogundare, Adekunle Bamidele Taiwo, Odunayo Adebukola Fatunla.

**Methodology:** Ezra Olatunde Ogundare.

**Resources:** Ezra Olatunde Ogundare.

**Supervision:** Ezra Olatunde Ogundare.

**Writing – original draft:** Ezra Olatunde Ogundare.

**Writing – review & editing:** Ezra Olatunde Ogundare, Adebukola Bidemi Ajite, Adewuyi Temidayo Adeniyi, Adefunke Olarinre Babatola, Adekunle Bamidele Taiwo, Odunayo Adebukola Fatunla, Alfred Airemionkhale, Oladele Simeon Olatunya, Oyeku Akibu Oyelami.

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
