## [Decision Letter · Decision Letter 0]

13 Oct 2021

Dear Dr. OGUNDARE,

Thank you very much for submitting your manuscript "NEONATAL TETANUS: IS NIGERIA TRULY MAKING PROGRESS? A TEN-YEAR REVIEW OF CASES MANAGED AT A TERTIARY HEALTH FACILITY IN SOUTHWEST NIIGERIA" for consideration at PLOS Neglected Tropical Diseases. As with all papers reviewed by the journal, your manuscript was reviewed by members of the editorial board and by several independent reviewers. In light of the reviews (below this email), we would like to invite the resubmission of a significantly-revised version that takes into account the reviewers' comments. 

We cannot make any decision about publication until we have seen the revised manuscript and your response to the reviewers' comments. Your revised manuscript is also likely to be sent to reviewers for further evaluation.

Sincerely,

Joseph M. Vinetz

Deputy Editor

Joseph Vinetz

Deputy Editor

Reviewer's Responses to Questions

**Key Review Criteria Required for Acceptance?**

**Methods**

-Are the objectives of the study clearly articulated with a clear testable hypothesis stated?

-Is the study design appropriate to address the stated objectives?

-Is the population clearly described and appropriate for the hypothesis being tested?

-Is the sample size sufficient to ensure adequate power to address the hypothesis being tested?

-Were correct statistical analysis used to support conclusions?

-Are there concerns about ethical or regulatory requirements being met?

Reviewer #1: What is the socioeconomic background of Ekiti, the catchment area of EKSUTH. This information is important given the fact that tetanus is a disease of socioeconomic relevance. Information should be provided on how tetanus is managed in the unit. Any management protocol in place? What facilities in the management of neonatal tetanus and its complications are available in the centre. How was autonomic dysfunction diagnosed and managed?

Reviewer #2: No Testable hypothesis

The study design is appropriate for the stated objective

The population is clearly described

No hypothesis is being tested (sample size issue not applicable)

The statistical method does not support the conclusions

Ethical and regulatory concerns adequately addressed

**Results**

-Does the analysis presented match the analysis plan?

-Are the results clearly and completely presented?

-Are the figures (Tables, Images) of sufficient quality for clarity?

Reviewer #1: Annual incidence rates should be plotted rather than the yearly distribution of absolute number of cases as shown in Figure 1. The incidence should be calculated as the number of cases in a year divided by the total admission for the year per 1000. That should be the emphasis in this study. The other aspects are at variance with the title of the manuscript and they add no new knowledge to the existing literature.

Reviewer #2: The analysis presented does not match the plan

The results are clearly presented but probably incomplete;

*All the babies were said to have umbilicus as the site without indicating how that was arrived at

*Table 1 - the aspect on cord care did not indicate if the methods of cord care were mutually exclusive

*Table 4 - What exactly is meant by age class

*What constitutes autonomic dysfunction was not shown, it was listed as a comorbidity whereas autonomic dysfunction is a manifestation of NNT

*What is referred to appropriate cord care was not defined

*About half were said to have had sepsis. How was sepsis established or ruled out?

*What the p value is about is unclear 

*The figure on trend was expressed in absolute numbers without showing what proportion it is to total admissions. This may not depict the true picture with respect to trend in prevalence

**Conclusions**

-Are the conclusions supported by the data presented?

-Are the limitations of analysis clearly described?

-Do the authors discuss how these data can be helpful to advance our understanding of the topic under study?

-Is public health relevance addressed?

Reviewer #1: The conclusions need to be re-written when the focus of the study is re-addressed as suggested.

Reviewer #2: The conclusions are not adequately supported by the data presented

The discussion needs to focus on the aim of the study

Several inferences were not derived from the data presented.

E.g. "It is surprising to note that none of the affected babies had chlorhexidine gel for cord care. This shows the need for increased awareness on approved cord care practices". The data did not show that lack of awareness was the reason for not using chlorhexidine.

Moreover, chlorhexidine cord care had not been adopted as national policy for half of the study period

3rd paragraph of discussion stated that 50% of cases presenting in the 1st week justifies the incubation period of 3days to 3weeks. That sentence is unclear

Page 17 - 1st line states that maternal ANC attendance was not associated with good outcome. The meaning of that is also unclear 

the total number of cases seen each year is too small to deduce that the prevalence is less especially as the total number of babies admitted annually is not indicated

**Editorial and Data Presentation Modifications?**

Reviewer #1: This reads like a general overview on neonatal tetanus and should be reduced to a third of the current length. Studies focused on trend of the disease should be reviewed to suggest the dwindling incidence across Africa and Asia. This should replace the details of risk factors and clinical presentation as currently presented. Specifically, what is the trend in tetanus toxoid coverage across the geopolitical zones in Nigeria and in Ekiti State in particular. This will provide information on the success or otherwise of the Maternal and Neonatal tetanus Elimination Initiative

The tetanus toxoid coverage over the preceding five years should be discussed as it is one of the contributory factors to the dwindling incidence of tetanus in the newborn. There are reports that Traditional Birth Homes and Spiritual Birth Homes (Churches) who run antenatal clinics now liase with primary health centres for tetanus toxoid administration during pregnancy. The practice of unhygienic cord care might also have reduced. Relevant literatures from Southwestern Nigeria should be sought.

Reviewer #2: (No Response)

**Summary and General Comments**

Reviewer #1: This is a retrospective study from a single centre in Nigeria. Remarkably, the cases presented are quite few and that made the statistical analysis bogus. The focus of the study is at variance with the title of the manuscript. The title suggests a trend analysis but the body of the manuscript dwells more on the clinical presentations of the disease. Going by the current title, a comparison of the incidence and case fatality rates at EKSUTH should be compared with similar data obtained from other centres across Nigeria over the same period of time. The data from a single centre will not be enough for generalization as the title suggests. Otherwise, the title should be amended to focus only on EKSUTH.

Reviewer #2: The topic is of public health importance.

I suggest modifications to the title as the study design and the result presented cannot answer the question posed in the title.

This is only a review of NNT in one hospital in one geopolitical zone of the country. Even the yearly trend for the hospital requires more information on the proportion relative to other causes of neonatal admissions which may be affected by several factors. it can therefore not be extrapolated to the whole country as the study was not powered for such. 

Introduction, paragraph 2 - other possible portal of entry of tetanus deserve to be mentioned 

Paragraph 3, 1st sentence in introduction should be referenced, same with paragraph 5

PLOS authors have the option to publish the peer review history of their article (what does this mean?). If published, this will include your full peer review and any attached files.

Reviewer #1: No

Reviewer #2: Yes: Olukemi O Tongo
---

## [Editor Report · Decision Letter 1]

19 Nov 2021

Dear Dr. OGUNDARE,

We are pleased to inform you that your manuscript 'A TEN-YEAR REVIEW OF NEONATAL TETANUS CASES MANAGED AT A TERTIARY HEALTH FACILITY IN A RESOURCE POOR SETTING: THE TREND, MANAGEMENT CHALLENGES AND OUTCOME' has been provisionally accepted for publication in PLOS Neglected Tropical Diseases.

Best regards,

Joseph M. Vinetz

Deputy Editor

Joseph Vinetz

Deputy Editor

---

## [Editor Report · Acceptance letter]

2 Dec 2021

Dear Dr. OGUNDARE,

We are delighted to inform you that your manuscript, "A TEN-YEAR REVIEW OF NEONATAL TETANUS CASES MANAGED AT A TERTIARY HEALTH FACILITY IN A RESOURCE POOR SETTING: THE TREND, MANAGEMENT CHALLENGES AND OUTCOME," has been formally accepted for publication in PLOS Neglected Tropical Diseases.

Best regards,

Shaden Kamhawi

co-Editor-in-Chief

Paul Brindley

co-Editor-in-Chief
